# Durability Assessment of Recycled Aggregate HVFA Concrete

**Valeria Corinaldesi** [1],*, **Jacopo Donnini** [2], **Chiara Giosué** [2], **Alessandra Mobili** [2] and **Francesca Tittarelli** [2]

1 Dipartimento di Scienze e Ingegneria della Materia, dell'Ambiente ed Urbanistica, Università Politecnica delle Marche, 60131 Ancona, Italy

2 Engineering Faculty, Università Politecnica delle Marche, 60131 Ancona, Italy; jacopo.donnini@staff.univpm.it (J.D.); c.giosue@staff.univpm.it (C.G.); a.mobili@staff.univpm.it (A.M.); f.tittarelli@univpm.it (F.T.)

* Correspondence: v.corinaldesi@staff.univpm.it; Tel.: +39-071-220-4428

**Abstract:** The possibility of producing high-volume fly ash (HVFA) recycled aggregate concrete represents an important step towards the development of sustainable building materials. In fact, there is a growing need to reduce the use of non-renewable natural resources and, at the same time, to valorize industrial by-products, such as fly ash, that would otherwise be sent to the landfill. The present experimental work investigates the physical and mechanical properties of concrete by replacing natural aggregates and cement with recycled aggregates and fly ash, respectively. First, the mechanical properties of four different mixtures have been analyzed and compared. Then, the effectiveness of recycled aggregate and fly ash on reducing carbonation and chloride penetration depth has been also evaluated. Finally, the corrosion behavior of the different concrete mixtures, reinforced with either bare or galvanized steel plates, has been evaluated. The results obtained show that high-volume fly ash (HVFA) recycled aggregate concrete can be produced without significative reduction in mechanical properties. Furthermore, the addition of high-volume fly ash and the total replacement of natural aggregates with recycled ones did not modify the corrosion behavior of embedded bare and galvanized steel reinforcement.

**Keywords:** fly ash; HVFA; recycled aggregate; RAC; sustainable building; reinforced concrete; corrosion of concrete

## 1. Introduction

In order to contribute to sustainable construction processes, some building materials, no longer able to fulfill their original task, can be reused as aggregate for concrete after being adequately processed [1]. Replacing natural aggregate (Nat) with recycled aggregate (Rec) in concrete allows the protection of the environment, since it reduces both the impact of quarries from which virgin aggregates are extracted and the volume of rubble disposed to landfills.

Similarly, the employment of fly ash (FA) in concrete enables the recycling of an industrial waste product. In particular, due to its pozzolanic activity, FA can partially replace cement, thus reducing the energy consumption and carbon dioxide emissions related to cement production [2].

Unlikely, replacing Nat with Rec can significantly reduce the performances of concrete in terms of workability. Moreover, Rec, due to its higher porosity with respect to Nat [3,4], also penalizes the concrete's compressive and tensile strength, the stiffness, the permeability, and the adherence between steel reinforcing bars and cement paste.

However, the literature has reported that the addition of mineral admixtures as fly ash (FA), metakaolin, silica fume, and ground granulated blast furnace slag in the mix is able to mitigate these

worsening effects both in traditional [5] and in self compacting concretes (SCC) [6]. In fact, generally, these additions seem able to improve more the properties of Rec concrete than those of Nat concrete [7].

Previous experiments [1,8] have already shown the feasibility of manufacturing structural concretes with Rec and high-volume fly ash (HVFA) since FA, by refining the pore structure, reduces the macro-pores volume. In this way, performance similar to Nat concrete can be achieved except for somewhat lower stiffness of the Rec mixture.

Water absorption [9], chloride ion penetration [10], sulphate attack [11], and shrinkage [12] increase with the increasing incorporation level of Rec. However, the addition of HVFA counteracts this effect [5] thanks to the chemical reaction between some particles of FA, that act as a pozzolanic addition instead of a filler, with Rec [12].

Wei et al. [13] have indicated that an adequate amount of Rec can even increase the frost resistance of concrete, especially when a low amount of FA is added, thanks to optimization of the concrete pore distribution [14].

Moreover, since the thermal expansion coefficient of the new cement paste is similar to that of the cement paste adhered to Rec, Rec concrete deteriorates less in terms of mechanical and durability properties than Nat after high temperature exposures [15], especially when FA is used as mineral admixture [16,17]. FA as bacteria immobilizer also improves the crack healing capacity of Rec concrete [18].

Concerning carbonation resistance, Rec and HVFA concrete suffer a deeper carbonation depth with respect to Nat concrete [19], also in SCC [20]. However, again, the incorporation of FA in Rec concrete allows the counteracting of this problem thanks to a synergistic effect between Rec and FA [9,21,22].

According to Limbachiya et al. [11], the best amount of coarse Rec in concrete is 30%, whereas up to 30% Rec does not significantly affect the concrete's properties. Regarding FA, European standards EN 197-1 and EN 206 limit the incorporation level of FA to 35% by cement mass, since at higher amounts, FA behave as a filler rather than as a binder. However, these two limits for Rec and FA can be exceeded in concrete mixes incorporating both FA and Rec [5]. After 90 days of curing, concretes manufactured with about 50% Rec and 50% FA can be generally classified at the same strength class of the control mix.

Rawaz Kurda et al. [22], thanks to a multicriteria decision method for concrete optimization (CONCRETOP), have shown that the best concrete mixes in terms of both concrete properties and cost and environmental impact, are those manufactured with both FA and Rec additions, rather than with only FA or Rec. In particular, the Global Warming Potential (GWP) of concrete mixes depends on the FA and Rec dosage ratio rather than the dosage of the single materials [23]. Moreover, the GWP of Rec strongly depends on the transportation scenario, but this effect significantly decreases with FA addition [24].

Therefore, it has been already widely proved that replacement of Nat with Rec and the replacement of cement with HVFA, given a little bit of compromise towards strength and durability aspect, can give great benefits to both economic and ecological aspects.

As reported above, many researchers have already studied the different properties of Rec concrete with HVFA. However, durability, which is a key property to ensure sustainable application of these materials in the construction sector, still needs more research to be fully investigated.

In this field, in particular, the literature still reports very few works on the protection offered by HVFA and Rec concrete to the corrosion of reinforcing bars. Stambaugh et al. [25], thanks to the theoretical development, validation, and implementation of a 1D numerical service-life prediction model for RCA, have affirmed that the use of either FA or slag allows the achievement of a 50-year service life for Rec concrete in chloride-laden environments. By the salt ponding test, Rehvati et al. [26] have stated that impermeable and high-quality Rec concrete, able to give high corrosion resistance to reinforcements, can be produced by replacing 20–30% cement with FA. In Gurdián et al. [27], no significant differences in the corrosion resistance of reinforced Rec concretes, manufactured with

15% of spent fluid catalytic cracking catalyst and 35% of FA, and Nat concretes under a natural chloride attack have been observed.

Moreover, in our knowledge, the corrosion behavior of galvanized steel reinforcements in reinforced RCA in HVFA has never been investigated.

Therefore, the purpose of this work is to determine whether the sustainability issue introduced in concrete design by Rec and HVFA would have any adverse effect on the durability of reinforced concrete in terms of penetration speed of chloride and carbon dioxide, and in terms of corrosion of bare or galvanized steel reinforcement embedded in concrete, if cracked.

To investigate the single and combined effect of Rec and HVFA addition on concrete properties, four different concrete mixes were prepared and compared:

1. Nat, as reference;
2. Nat + FA with HVFA added at equal amounts as cement;
3. Rec, by replacing the 100% of Nat with Rec;
4. Rec + FA with HVFA.

The different mixtures were compared in terms of mechanical performances, carbonation and chlorides penetration, and corrosion behavior of embedded bare and galvanized steel reinforcements.

## 2. Materials and Methods

### 2.1. Materials

As cementitious binder, Portland limestone blended cement type CEM II/A-L 42.5 R was used. The cement's Blaine fineness was 0.418 $m^2$/g and its density was 3.04 kg/$m^3$. The cement's chemical composition is reported in Table 1.

**Table 1.** Chemical composition of cement and fly ash.

| Oxide (%) | CEM II/A-L 42.5 R | Fly Ash |
|---|---|---|
| $SiO_2$ | 29.7 | 59.9 |
| $Al_2O_3$ | 3.7 | 22.9 |
| $Fe_2O_3$ | 1.8 | 4.7 |
| $TiO_2$ | 0.1 | 0.9 |
| CaO | 59.3 | 3.1 |
| MgO | 1.1 | 1.6 |
| $SO_3$ | 3.2 | 0.3 |
| $K_2O$ | 0.8 | 2.2 |
| $Na_2O$ | 0.3 | 0.6 |
| Loss on Ignition (L.o.I.) | 11.6 | 3.3 |

Two virgin aggregate fractions were used: limestone aggregate from a quarry (up to 15 mm particle size) and quartz sand (up to 6 mm particle size). Their grain size distribution is shown in Figure 1 and their physical properties are reported in Table 2.

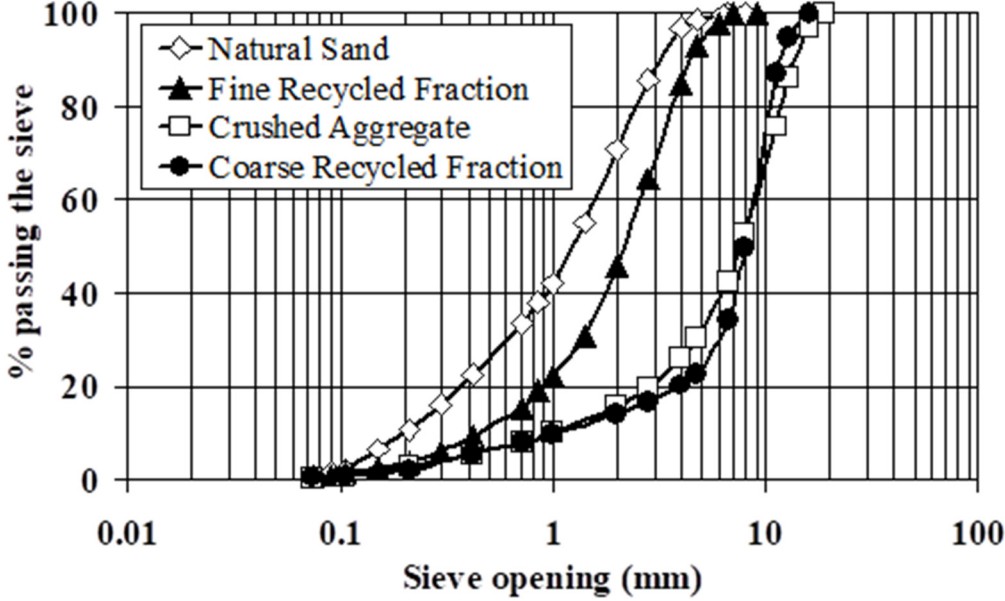

**Figure 1.** Grain size distribution curves of the aggregate fractions.

**Table 2.** Physical properties of the aggregate fractions.

| Aggregate Fractions | Bulk Density (SSD), kg/m$^3$ | Water Absorption, % | Passing 75 μm, % |
|---|---|---|---|
| Natural Sand | 2620 | 3 | 0.9 |
| Crushed Aggregate | 2680 | 2 | 0.2 |
| Fine Recycled Fraction | 2150 | 10 | 0.5 |
| Coarse Recycled Fraction | 2320 | 8 | 0.3 |

Two recycled aggregate fractions were used: coarse aggregate (up to 15 mm) and fine aggregate (up to 6 mm). The origin of these recycled aggregates was a recycling plant in Villa Musone (Italy), where debris coming from building demolition has been selected, crushed, cleaned, and finally, sieved. The grain size distribution of recycled aggregates is shown in Figure 1 and their physical properties are reported in Table 2.

As water-reducing admixture, an acrylic-based superplasticizer (in the form of 30% aqueous solution) was added, when required, to reach the optimal workability degree.

Low calcium fly ash (according to the definition of ASTM C 618 Class F) coming from a thermal power plant located in La Spezia (Italy), with Blaine fineness of 0.458 m$^2$/g and density of 2.23 kg/m$^3$, was used. Fly ash chemical composition is detailed in Table 1.

*2.2. Mixture Proportions*

Four different concrete mixtures were designed as reported in Table 3. The two different kinds of aggregate particles, either recycled or natural, were used by maintaining the same grain size distribution (up to 15 mm). The optimization of grain size particle distribution was achieved by suitable combining of fine and coarse aggregate fractions, according to Bolomey [12]. All mixtures were designed so to have the same workability, with a slump value in the range 150–180 mm. According to that, when recycled aggregates and fly ash have been used, an acrylic-based superplasticizer was added at dosages up to 2.0% by weight of cement.

**Table 3.** Concrete mixture proportions (kg/m$^3$).

|  | Nat-0.6 | Nat + FA-0.6 | Rec-0.3 | Rec + FA-0.6 |
|---|---|---|---|---|
| Water/Cement | 0.6 | 0.6 | 0.3 | 0.6 |
| Water/Binder | 0.6 | 0.3 | 0.3 | 0.3 |
| Water | 250 | 250 | 180 | 180 |
| Cement | 420 | 420 | 600 | 300 |
| Fly ash | - | 420 | - | 300 |
| Superplasticizer | - | 8.4 | 6.0 | 5.4 |
| Natural sand | 290 | - | - | - |
| Fine recycled aggregate | - | - | 125 | 125 |
| Crushed aggregate | 1280 | 1280 | - | - |
| Coarse recycled aggregate | - | - | 1030 | 1030 |

The first reference mixture (Nat-0.6) was designed with only the addition of virgin aggregates and a water-to-cement ratio (*w/c*) of 0.60.

In the second mixture (Nat + FA-0.6), fly ash (FA) at the same dosage of cement was added, in replacement of natural sand. However, to reach the same workability of the reference mixture, acrylic-based superplasticizer as water-reducing admixture was added at a dosage of 2.0% by weight of cement.

In the third concrete mixture (Rec-0.3), virgin aggregates were fully substituted with recycled ones, while a lower *w/c* equal to 0.3 was adopted, to recover the strength loss due to the addition of weaker aggregate.

Finally, in the fourth mixture (Rec + FA-0.6), fine and coarse recycled aggregates were used in complete substitution of natural aggregates, while cement was partially replaced with fly ash. The use of superplasticizer at dosage of 1.8% by weight of cement allowed us to keep unchanged the water-to-cement ratio (equal to 0.6).

## 3. Preparation and Testing of Specimens

### 3.1. Compression Tests

Compression tests were performed on cubic specimens (100 mm edge) after 3, 7, 28, and 56 days of wet curing at 20 °C (according to the procedure of UNI EN 12390-1 [28]). Compression tests were carried out according to the procedure described in the Italian Standards UNI EN 12390-3 [29]. Three specimens for each curing time and each type of mixture were tested.

### 3.2. Carbonation Depth

Carbonation depth was evaluated through a phenolphthalein test (following the indication reported in RILEM CPC-18 [30]) on three cubic concrete specimens (100 mm edge) for each mixture, exposed to the open air at an average temperature of 20 °C (only for the first day of curing, wet curing was adopted).

### 3.3. Chloride Penetration

Chloride penetration speed into concrete was evaluated by means of silver nitrate and fluorescein test [13]. Both solutions were sprayed on the two cracked surfaces obtained by splitting the cubic (100 mm edge) concrete specimens. These specimens were previously wet-cured for 7 days, air-cured for 21 days at a temperature of 20 °C, and finally, exposed to 10% sodium chloride aqueous solution.

*3.4. Corrosion Tests*

Concrete specimens with dimensions of $280 \times 70 \times 70$ mm$^3$ were manufactured for electrochemical tests. Each specimen was reinforced with a bare or galvanized steel plate ($210 \times 40 \times 1$ mm$^3$), embedded within a 3 cm concrete cover. Steel plates were used instead of common bars because they can allow specimen cracking without splitting and they offer a higher anodic area at the crack apex. The galvanized steel plates, obtained by molten zinc immersion, were covered by a 100 μm thick zinc layer, with an outer pure zinc layer (η phase) about 20 μm thick. The galvanized reinforcements, just before being embedded in the fresh concrete, were submerged for 5 s in a 15% sodium hydroxide solution to dissolve the $ZnCO_3$ layer formed during air exposure. The electric contacts between the metallic plates and the measurement equipment were arranged as reported in [14].

After 1 month of air curing at T = 20 ± 3 °C and RH = 50 ± 5%, a crack width of 1 mm was produced in a pre-formed notch area of the specimens by applying a flexural stress with the apex crack reaching the metallic plates. Then, the specimens were exposed to weekly wet–dry cycles (2 days dry and 5 days wet) in a 10% NaCl solution.

The corrosion risk of the reinforcement in the concrete specimens exposed to the chloride environment was evaluated by corrosion potential measurements by using a saturated calomel electrode (SCE) as a reference. The kinetics of the corrosion process was followed by polarization resistance measurements through the galvanodynamic method, where an external graphite bar was used as a counter-electrode. The polarization resistance was calculated as the average value between the anode and the cathode branch.

In the following graphs, the reported electrochemical values are averages of the measurements carried out on 3 specimens of each type during the full immersion period.

In order to validate the electrochemical tests, after 7 wet–dry cycles in the chloride solution, the concrete specimens were saw-cut and all the metallic plates were removed after splitting the concrete specimens, to evaluate the reinforcement corrosion by visual observation. The surface of the corroded area on bare steel plates was evaluated after pickling, whereas metallographic analysis was carried out on the cross-section of the galvanized steel plates to evaluate the coating thickness decrease due to the corrosive attack.

## 4. Results and Discussion

*4.1. Compression Tests*

The experimental results of compressive tests on the four mixtures at different curing times are reported in Figure 2. All mixtures showed a compressive strength greater than 27 MPa at 28 days. At early ages (3 and 7 days), the different *w/c* influenced the compressive strength more than the kind of aggregate. The total substitution of natural aggregates with recycled ones, simultaneously with the reduction in *w/c* and the use of superplasticizer (Rec-0.3), did not substantially modify the compressive strength with respect to the concrete mixture with natural aggregates (Nat-0.6). When cement was partially replaced with FA (Rec + FA-0.6), the mixture showed a lower compressive strength at early ages (3 and 7 days) and a slightly higher compressive strength after 56 days of curing, thanks to the fly ash pozzolanic activity which develops at long ages. The use of lighter aggregate (i.e., recycled aggregate) increasingly influenced the compressive strength value as the cement matrix became stronger, since it represents the "weak link" in the chain [15]. The best results have been obtained for the mixture realized with natural aggregates and fly ash (Nat + FA-0.6), which showed a compressive strength of about 44 MPa after 56 days of curing.

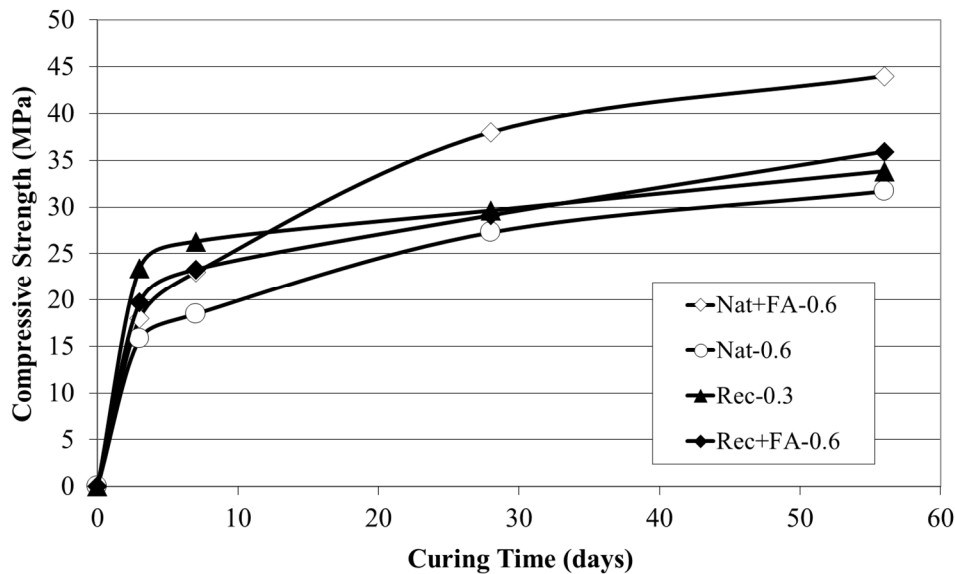

**Figure 2.** Compressive strength vs. curing time.

### 4.2. Carbonation Depth

In Figure 3, the measured values of carbonation depth ($x$, in mm) were reported vs. days of air exposure ($t$). Experimental data can be quite accurately described by a linear relationship between the carbonation depth and square root of time (following the law $x = k \cdot \sqrt{t}$). The higher carbonation depth, equal to about 7 mm after 1 year of exposure, was found for the mixture with natural aggregates (Nat-0.6), while the lower carbonation depth, equal to 2.8 mm, was found in the concrete mixture realized with natural aggregates and fly ash. These results confirm that the addition of high volumes of fly ash is able to significantly reduce the carbonation process, even when a porous aggregate (such as the recycled aggregate) is used, thanks to the refinement of the pores and a consequent improved microstructure.

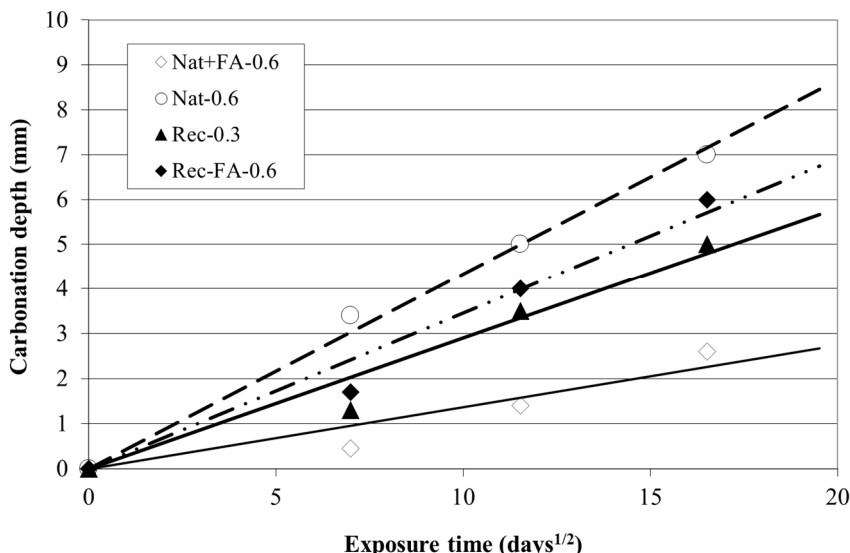

**Figure 3.** Carbonation depth vs. time of air exposure.

### 4.3. Chloride Penetration

In Figure 4, chloride penetration depth values are reported vs. time of exposure to 10% NaCl aqueous solution after saturation with water of concrete specimens. Figure 4 shows only data obtained

after about 28-day water immersion due to initial instability phenomena (in practice, the chloride binding with cement matrix can interfere with the chloride migration [16]).

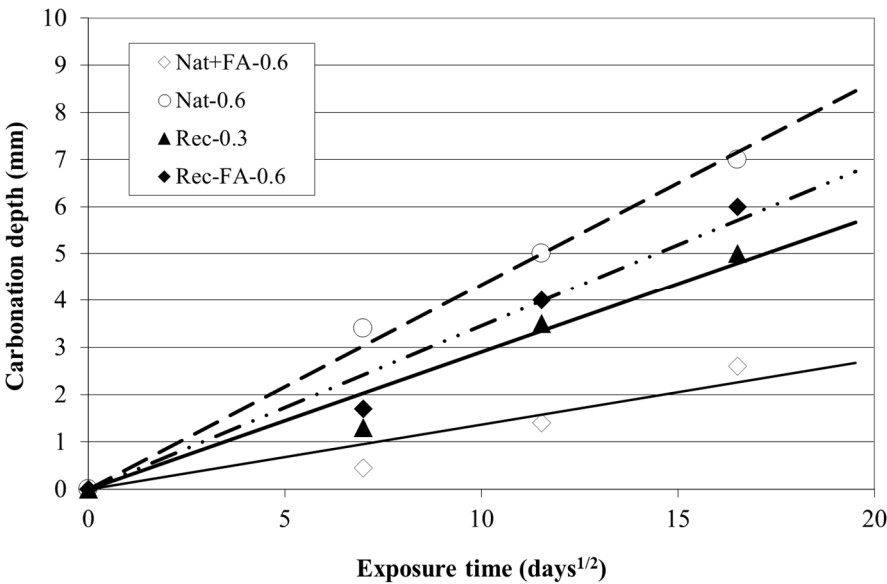

**Figure 4.** Chloride penetration depth vs. immersion time.

Collepardi et al. [13] demonstrated that chloride penetration depth ($x$) varies with immersion time ($t$) according to Equation (1), which can be obtained from the solution of Fick's second law under non-steady-state conditions for diffusion in a semi-finite solid:

$$x = 4 \cdot \sqrt{D \cdot t} \tag{1}$$

where $D$ is the diffusion coefficient of Cl ions into wetted concrete pores, expressed as $cm^2 \cdot s^{-1} \cdot 10^{-8}$. The $D$ values, coming from (1) by interpolation of the results reported in Figure 4, have been calculated (see Table 4). It can be noticed a significant beneficial effect due to fly ash addition on the chloride penetration depth, which has been measured for Rec + FA-0.6 and Nat + FA-0.6 concretes, respectively. As a matter of fact, Cl diffusion coefficients into these mixtures are 10 times less than those measured for the other concrete mixtures.

**Table 4.** Chloride diffusion coefficients at 20 °C.

|  | Nat-0.6 | Nat + FA-0.6 | Rec-0.3 | Rec + FA-0.6 |
|---|---|---|---|---|
| D ($cm^2 \cdot s^{-1} \cdot 10^{-6}$) | 1.90 | 0.87 | 0.72 | 0.46 |

Collepardi et al. [13] found experimental results, which support the hypothesis that the different structure of pore surfaces created with the addition of pozzolanic materials played an important role in influencing concrete porosity while chloride ions penetrate it. In fact, if concrete is prepared by adding fly ash, chloride binding operated by the cement matrix significantly increases.

A lower water/cement resulted beneficial in terms of hindering of chloride penetration. In fact, the Cl diffusion coefficient into the Nat-0.6 mixture is double than Rec-0.3, even if recycled instead of virgin aggregate was employed. Nevertheless, aggregate pore structure could have a significant effect on concrete permeability to chloride penetration (as you can see by comparing Nat + FA-0.6 and Rec + FA-0.6 mixtures in Figure 4). Zhang et al. [17] experimented several lightweight aggregate concretes and the conclusion was that concrete permeability depends more on cement matrix porosity than lightweight aggregate porosity.

### 4.4. Corrosion Tests

4.4.1. Bare Steel Plates

Figures 5 and 6 show, respectively, the free corrosion potential values and the corrosion rates of bare steel plates embedded in cracked concrete as a function of wet–dry cycles. Just after the exposure to the chloride environment, all the steel plates assumed activation values lower than −500 mV/SCE, reflecting a general high corrosion risk, regardless of the type of concrete mixture. At the same time, the related polarization resistance values did not change significantly with the addition of FA or when natural aggregates were replaced with recycled ones, thus indicating similar corrosion rates for the different concretes. Therefore, the total replacement of natural aggregates, with or without high FA volume, does not seem to negatively affect the corrosion behavior of embedded steel reinforcements when an adequate strength class is guaranteed. Moreover, despite the reduction in concrete pore solution alkalinity due to the pozzolanic reaction of fly ash, the corrosion behavior of steel reinforcements in high-volume fly ash concrete seems not to be negatively affected, at least for cracked concrete.

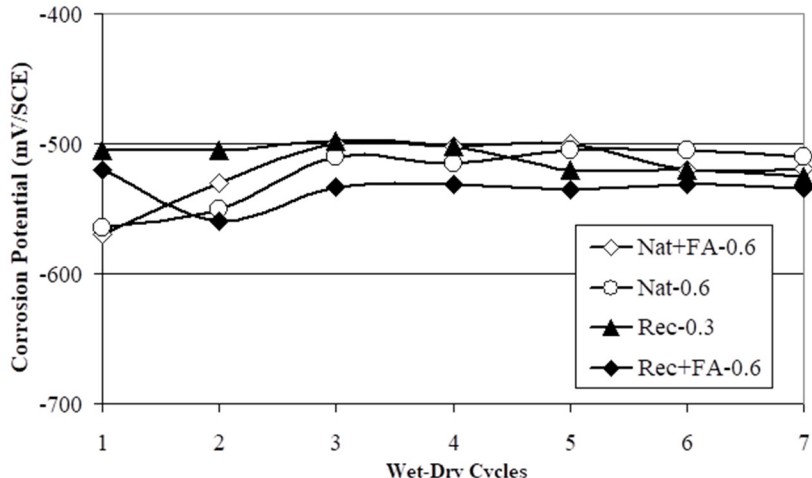

**Figure 5.** Corrosion potential of bare steel plates in cracked concrete as a function of wet–dry cycles.

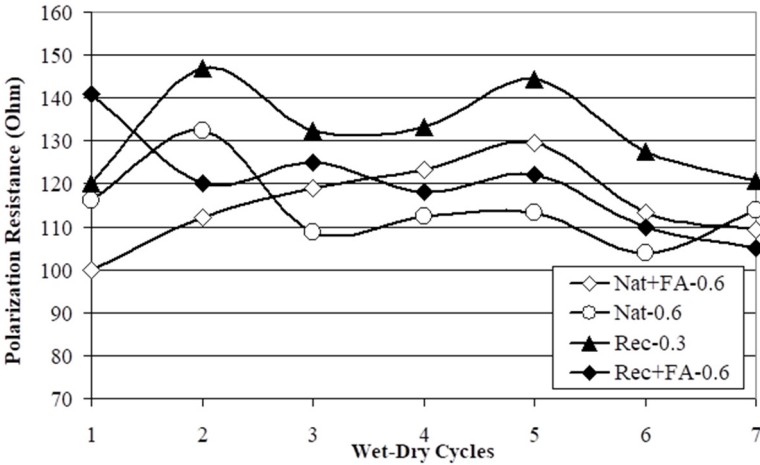

**Figure 6.** Polarization resistance of bare steel plates in cracked concrete as a function of wet–dry cycles.

To better analyze the electrochemical behavior, all specimens were autopsied after 7 wet–dry cycles in order to visually evaluate the corroded area and to assess the weight loss of the bare steel plates after pickling. A visual observation of the corrosive attack on the steel plates embedded in the different concrete mixtures is reported in Figure 7. All the bare steel plates showed significant

corrosive attack in the crack area, due to the high Cl concentration detected on the steel reinforcement surface (2–5% by weight of cement), thus overcoming the concentration threshold (0.4% by cement weight), which is considered the critical value able to induce the corrosion of bare steel. However, from a morphological point of view, in the presence of recycled aggregate (Rec-0.3) or when a high amount of fly ash is used (Nat + FA-0.6), the corrosive attack appears more diffuse and less penetrating (see Figure 7).

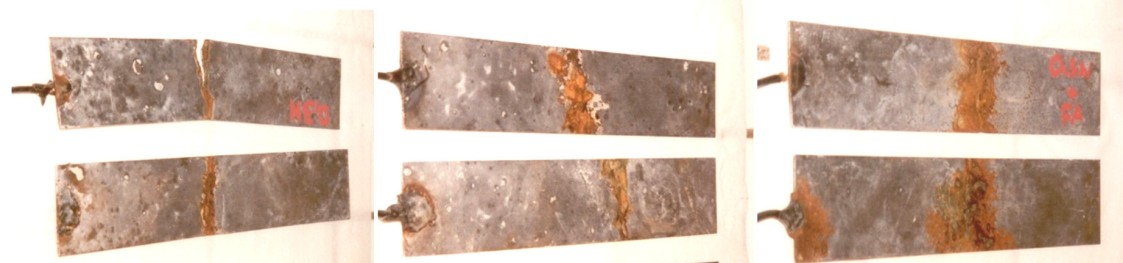

**Figure 7.** Visual observation of the corrosive attack on the bare steel plates in reference natural aggregate concrete (Nat-0.6, on the left), in recycled aggregate concrete (Rec-0.3, in the middle), and high-volume fly ash concrete (Nat + FA-0.6, on the right).

### 4.4.2. Galvanized Steel Plates

The free corrosion potential and the polarization resistance of galvanized steel plates embedded in cracked concrete, as a function of wet–dry cycles, are reported in Figures 8 and 9, respectively. The electrochemical tests showed no significant increase in the corrosion rate of galvanized steel embedded in concrete mixtures containing recycled aggregate and/or FA. Indeed, the corrosion risk seems to be even lower in recycled aggregate concrete (see Rec-0.3 in Figure 8) than in the other mixtures.

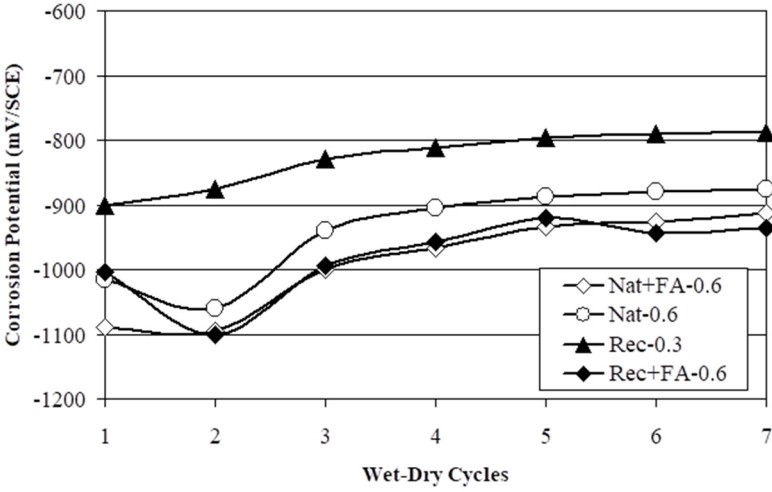

**Figure 8.** Corrosion potential of galvanized steel plates in cracked concrete vs. wet–dry cycles.

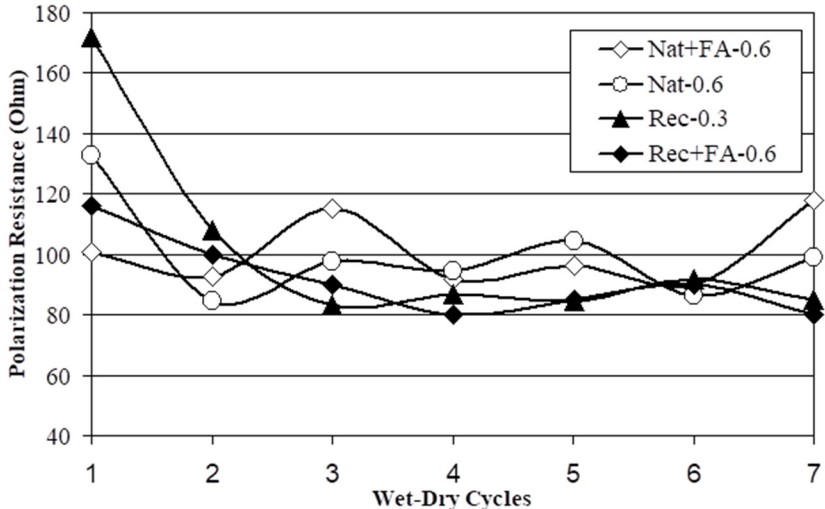

**Figure 9.** Polarization resistance of galvanized steel plates in cracked concrete vs. wet–dry cycles.

The visual observation of the interface between galvanized steel plates and concrete added unexpected information to that obtained by the electrochemical tests. Indeed, in the concrete without additions (Nat-0.6, Figure 10a), far from the crack apex, a Fe-Zn alloy appeared on the surface of the reinforcement, meaning total consumption of the η zinc layer due to the corrosive attack, as later confirmed by metallographic observation (Figure 10b). On the other hand, the galvanized steel plates extracted from recycled aggregate concrete (Rec-0.3) showed a less deep corrosive attack. Zinc grains were still visible on the galvanized surface after exposure to wet–dry cycles (Figure 11a). Moreover, the metallographic observation revealed that a continuous thick η zinc layer was still present on the metallic plate (Figure 11b). Similar results were observed when high-volume fly ash was added in the mix.

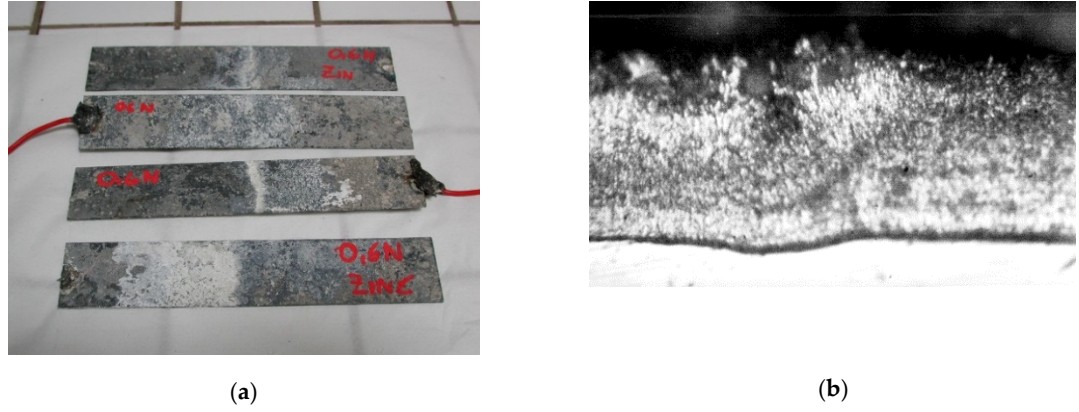

(**a**)                                                          (**b**)

**Figure 10.** Visual obs. (**a**) and metallographic cross section (**b**) of galvanized steel plate in Nat-0.6

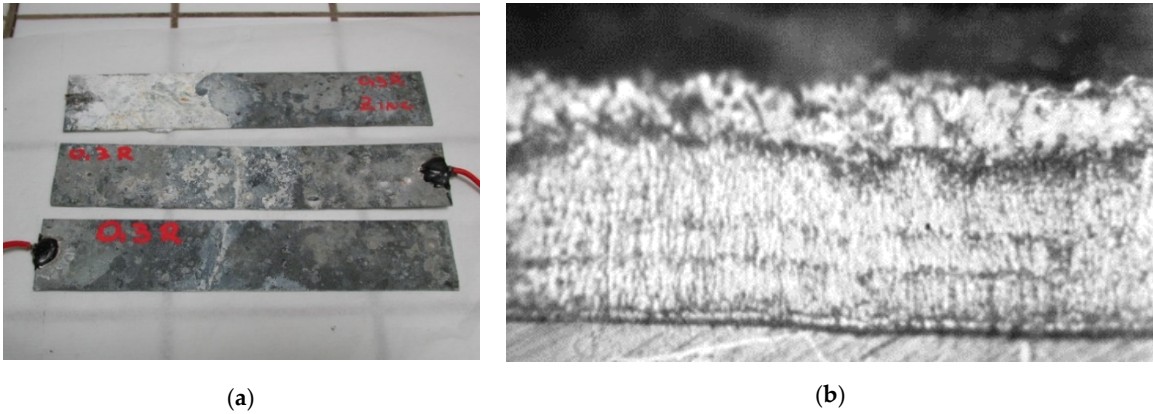

| (**a**) | (**b**) |

**Figure 11.** Visual obs. (**a**) and metallographic cross section (**b**) of a galvanized steel plate in Rec-0.3

## 5. Conclusions

Based on the experimental investigation carried out on recycled aggregate HVFA concrete to study its mechanical properties and durability characteristics, the following conclusions can be drawn:

- The use of recycled aggregate fractions (up to 15 mm) proved to be suitable for manufacturing structural concrete, even when employed in total replacement of fine and coarse natural aggregates.
- If natural aggregates are substituted by recycled ones, fly ash is added to the mixture in partial substitution of cement, and the *w/c* is decreased with the aid of a superplasticizer, the compressive strength is equal or even greater than that of natural aggregate concrete.
- The use of fly ash proved to be very effective in reducing carbonation and Cl penetration depths in concrete, even if recycled instead of virgin aggregate was used.
- When the concrete cover was cracked, the addition of fly ash and the use of recycled aggregates (Rec + FA-0.6) did not reduced the corrosion resistance of steel reinforcement, while it appeared very effective to protect the galvanized steel reinforcement.
- An improved attention to the sustainability issue in concrete manufacturing, promoted by using recycled aggregate and high-volume fly ash, did not cause bad side effects on durability performance of reinforced concrete specimens under testing.

**Author Contributions:** Investigation, V.C., J.D., C.G., A.M. and F.T.; Methodology, J.D. and F.T.; Supervision, V.C.; Writing—review and editing, J.D. and F.T. All authors have read and agreed to the published version of the manuscript.

**Funding:** This research received no external funding.

**Conflicts of Interest:** The authors declare no conflict of interest.

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
