# Peer review of "Durability Assessment of Recycled Aggregate HVFA Concrete"

_applsci, doi:10.3390/app10186454_

Round 1
Reviewer 1 Report
Title: Durability Assessment of Recycled Aggregate HVFA Concrete
The paper investigates the effectiveness of recycled aggregate and fly ash on reducing carbonation and chloride penetration depth, as well as the effect on the corrosion behavior of either bare or galvanized steel in cracked reinforced concrete. After reviewing this paper, some comments and suggestions on this paper are indicated as follows:
- There are several published papers in terms of using high-volume fly ash in recycled aggregate concrete which is not new and this topic has been investigated widely in the previous studies. So, it is not clear what makes the present study unique. What is the gap of literature, what is the main contribution of the study, which is different from others should be clearly clarified? It needs to be clearly stated the contributions of the manuscript in the introduction section. Although the study conducted many experimental tests, the experimental results did not make the manuscript novelty.
- The abstract needs to be revised and provided the important results from experiments or the main contribution of the manuscript.
- Explain the reasons for the experimental program and concrete mix proportion.
- Discussion or explanations are too speculative and not convict the readers. Please provide suitable reasons and more discussion on the results that have been mentioned in the manuscript. The authors may also provide the mechanism and theories for explanations.
- The manuscript should provide the deviation of all result in all figures
- Conclusions section, the authors cannot suggest these conclusions based on the experimental campaign in this manuscript. These conclusions are not new compared with the previous conclusions. Please provide the main contribution of the study.
- I suggest measuring more mechanical properties of recycled aggregate concrete containing high-volume fly ash.
Author Response
The paper investigates the effectiveness of recycled aggregate and fly ash on reducing carbonation and chloride penetration depth, as well as the effect on the corrosion behavior of either bare or galvanized steel in cracked reinforced concrete. After reviewing this paper, some comments and suggestions on this paper are indicated as follows:
There are several published papers in terms of using high-volume fly ash in recycled aggregate concrete which is not new and this topic has been investigated widely in the previous studies. So, it is not clear what makes the present study unique. What is the gap of literature, what is the main contribution of the study, which is different from others should be clearly clarified? It needs to be clearly stated the contributions of the manuscript in the introduction section. Although the study conducted many experimental tests, the experimental results did not make the manuscript novelty.
The abstract needs to be revised and provided the important results from experiments or the main contribution of the manuscript.
The abstract has been revised in order to provide the main information on the experimental tests carried out.
Explain the reasons for the experimental program and concrete mix proportion.
The experimental program was aimed at evaluating the influence of recycled aggregate and fly ash additions to the mechanical and durability performance of concrete. For this reason the two kinds of additions have been tested singularly (Nat+FA-0.6 and Rec-0.3) and coupled (Rec+FA-0.6).
The choice of a lower water to cement ratio for the mixture with recycled aggregate has been made for compensating the reduced strength expected.
Discussion or explanations are too speculative and not convict the readers. Please provide suitable reasons and more discussion on the results that have been mentioned in the manuscript. The authors may also provide the mechanism and theories for explanations.
Discussion of results has been revised and expanded.
The manuscript should provide the deviation of all result in all figures
Conclusions section, the authors cannot suggest these conclusions based on the experimental campaign in this manuscript. These conclusions are not new compared with the previous conclusions. Please provide the main contribution of the study.
Conclusions have been modified by providing the main contribution of this experimental study.
I suggest measuring more mechanical properties of recycled aggregate concrete containing high-volume fly ash.

Reviewer 2 Report
The introduction should be rebuilt, it is difficult to understand and unreadable with a lot of short sentences and the all letter abbreviations. The experiments are interesting but I have not found how many specimens of each mixture have you tested. The discussion of your results is chaotic and it is not easy to follow your conclusions. The final conclusion is to general for such an experimental work.
Author Response
The introduction should be rebuilt, it is difficult to understand and unreadable with a lot of short sentences and the all letter abbreviations. The experiments are interesting but I have not found how many specimens of each mixture have you tested.
The number of specimens tested for each mixture has been clearly explained in the revised manuscript.
The discussion of your results is chaotic and it is not easy to follow your conclusions. The final conclusion is to general for such an experimental work.
The final conclusion has been modified by providing the main contribution of this experimental study.

Reviewer 3 Report
The manuscript presents an experimental work aimed at using RA & FA to make concrete. There are following questions:
- Abstract need to be rewritten to report about the main and new findings obtained in this paper briefly, not just a general description.
- The results and discussion section of the manuscript must be improved significantly. It has been presented as a direct report, readers would like to know the reasons.
- Error bars should be added to all of the figures.
- The author should analyze the interaction between variables.
- Conclusion is too short.
Author Response
The manuscript presents an experimental work aimed at using RA & FA to make concrete. There are following questions:
Abstract need to be rewritten to report about the main and new findings obtained in this paper briefly, not just a general description.
Abstract has been completely revised by adding the main findings of this study.
The results and discussion section of the manuscript must be improved significantly. It has been presented as a direct report, readers would like to know the reasons.
The manuscript has been deeply revised and discussion of results improved.
Error bars should be added to all of the figures. The author should analyze the interaction between variables. Conclusion is too short.
Conclusions have been completely revised with reference to the experimental results.

Reviewer 4 Report
Dear authors,
Your research is very interesting. I only propose the following recommendations:
In line 69 I would suggest "According to Limbachiya et al [22] ...." instead of "According to [22] ..." also in line 94 "Revathi et al by ..."
Line 170 UNI 6130-72 Part II should appear in References. The same in line 175 RILEM CPC-18.
Finally, the conclusions are very brief, they should be expanded according to the results obtained.
Best regards
Author Response
Dear authors,
Your research is very interesting. I only propose the following recommendations:
In line 69 I would suggest "According to Limbachiya et al [22] ...." instead of "According to [22] ..." also in line 94 "Revathi et al by ..."
Thanks for your suggestion, the text has been modified.
Line 170 UNI 6130-72 Part II should appear in References. The same in line 175 RILEM CPC-18.
Modified.
Finally, the conclusions are very brief, they should be expanded according to the results obtained.
Conclusions have been completely revised with reference to the experimental results.

Round 2
Reviewer 1 Report
the Authors revised the manuscript carefully and sufficiently. please consider for publication.
Reviewer 3 Report
My Overall Recommendation: Accept.